# Renal Rehabilitation—Its Theory and Clinical Application to Patients Undergoing Daily Dialysis Therapy

**Ryota Matsuzawa [1],* and Daisuke Kakita [2]**

1  Department of Physical Therapy, School of Rehabilitation, Hyogo Medical University, Kobe 650-8530, Japan
2  Course of Health Science, Hyogo Medical University Graduate School of Health Science, Kobe 650-8530, Japan
*  Correspondence: ryota122560@gmail.com; Tel.: +81-78-304-3181; Fax: +81-78-304-2811

**Abstract:** An aging population and the prevalence of lifestyle-related ailments have led to a world-wide increase in the rate of chronic kidney disease requiring renal replacement therapy. The mean age of people requiring dialysis has been rising, and Japanese patients are aging more rapidly than those in the United States and Europe. Compared to people with normal kidney function, those undergoing hemodialysis are at increased risk of sarcopenia or frailty and serious health problems that limit access to kidney transplantation and lead to adverse health outcomes such as functional dependence, hospitalization, and death in patients on dialysis treatment. The Japanese Society of Renal Rehabilitation, established in 2011, published a clinical practice guideline for renal rehabilitation in 2019. Although the concept has become widely known among kidney health providers in recent years, efforts have still not focused on routine clinical care for patients with chronic kidney disease. In this review, the theory and clinical application of renal rehabilitation for patients undergoing daily hemodialysis were investigated.

**Keywords:** sarcopenia; frailty; exercise; CKD; dialysis

## 1. Introduction

As the world's population ages, the rate of chronic kidney disease requiring renal replacement therapy is increasing [1], along with the mean age of patients undergoing dialysis. Significant increases in the age of these patients is the result of improved survival and reduced transplant availability, which has been observed in almost all 12 nations in the international cohort "Dialysis Outcomes and Practice Patterns Study (DOPPS)" [2]. With previously reported data, Japanese patients on dialysis are aging more rapidly than those in the United States or Europe [3]. The rate of patients aged 65 years and older in the United States rose to 38.4% from 21.1% between 1980 and 2015, meanwhile, the rate in Japan rose to 59.8% from 12.2%. According to the nationwide data of the Japanese Society for Dialysis Therapy, the mean age of the Japanese dialysis population was 68.8 years at the end of 2018, a 14.22-year increase since the end of 1990. Furthermore, patients aged 60 years or older represented 79.1% of those who started dialysis therapy in 2018 and 78.1% of the entire dialysis population [4].

The health management of older hemodialysis patients poses serious issues that are not only clinical but also social. In 2016, the advisory board of European Renal Best Practice published guidelines on the management of older patients with chronic kidney disease [5]. In 2019, the Japanese Society of Renal Rehabilitation, which was established in 2011, also published a clinical practice guide for renal rehabilitation, targeting patients who were and were not dependent on dialysis and had a renal transplant [6]. In recent years, the concept of renal rehabilitation has gradually become widely known among kidney health providers, but it is currently still not included in routine clinical care for patients undergoing dialysis because of the lack of medical staff who can evaluate physical functions and provide exercise guidance adequately. This review focused on the theory of renal rehabilitation and its clinical application for patients on daily dialysis.

## 2. Functional Status and Physical Frailty

The mortality rate for patients on hemodialysis was approximately 10% in 2018 [4] and is still high despite continued improvements in dialysis technology. One of the potential contributors to poor survival might be a high burden of functional dependence [7], which is an individual's inability to perform day-to-day tasks associated with personal care and maintaining a household. A previous study that included almost all 12 nations demonstrated a high level of disability in daily activities in most patients undergoing hemodialysis, and a dose–response association was noted between poor functional status and adverse clinical outcomes [8]. Furthermore, the association between a yearly change in functional status and all-cause mortality among 817 Japanese individuals requiring hemodialysis therapy was examined [9]. Among the patients free of disability at the baseline, 19.9% experienced a functional decline during the one-year observation, which was strongly associated with a higher mortality risk. Importantly, even after adjusting for baseline characteristics including functional status, the reduction still had a negative effect on survival in patients with end-stage renal disease. We underscore the importance of regular monitoring of a patient's functional status and interventions to prevent deterioration. Impaired mobility, poor physical functioning, and muscle weakness, the main components of physical frailty, contribute to an increased likelihood of disabilities not only in community-dwelling older adults [10,11] but also in patients on hemodialysis [12].

Frailty is generally considered to be an age-related fragile state characterized by physiological vulnerability to stress, associated with an increased risk of adverse health outcomes [13,14]. The frailty phenotype was first defined by Fried and colleagues based on the following five criteria: shrinking, weakness, poor endurance and energy, slowness, and a low level of physical activity [13]. Frailty is identified by the presence of three or more of the above criteria, and an intermediate frailty phenotype is commonly defined as having one or two. Satake et al. proposed a revised Japanese version of the Cardiovascular Health Study (J-CHS) criteria (Table 1) [15], which was constructed by simplifying the original CHS criteria to suit older Japanese people.

**Table 1.** The revised Japanese version of the Cardiovascular Health Study (J-CHS) criteria, created based on a reference from Satake et al. [15].

| Component | Questions and Measurements | Answer |
|---|---|---|
| Shrinking | Have you unintentionally lost 2 or more kg in the past 6 months? | Yes = 1 No = 0 |
| Weakness | Grip strength <28 kg in men or 18 kg in women | Yes = 1 No = 0 |
| Exhaustion | In the past 2 weeks, have you felt tired without reason? | Yes = 1 No = 0 |
| Slowness | Gait speed <1.0 m/s | Yes = 1 No = 0 |
| Low activity | Do you engage in a moderate level of physical exercise or sports? Do you engage in a low level of physical exercise aimed at health? | No to both questions = 1 Other = 0 |

Frailty, prefrailty, and robustness were defined as 3–5, 1–2, and 0 points, respectively.

Frailty is highly prevalent in patients with chronic kidney disease who require hemodialysis. The prevalence of frailty in this population was 36.8% based on a previously performed meta-analysis [16] compared with 7.4% of community-dwelling older adults [17]. Many factors were mutually connected and could be unified theoretically into a cycle of frailty, which we have reported elsewhere (Figure 1) [18]. Patients with kidney diseases also have an increased risk of sarcopenia, which is defined as a state with low muscle mass and low muscle strength or low physical performance, and it is thought that sarcopenia occurs due to a comorbidity burden, long-standing malnutrition, chronic inflammation, metabolic acidosis, anabolic

resistance, hormonal changes, physical inactivity, and amino acid loss via dialysis [19–22]. Sarcopenia and physical frailty decrease access to kidney transplantation [23] and lead to adverse health outcomes including hospitalization [24] and death [25] in patients on hemodialysis. Thus, early identification and treatment are especially needed for these populations.

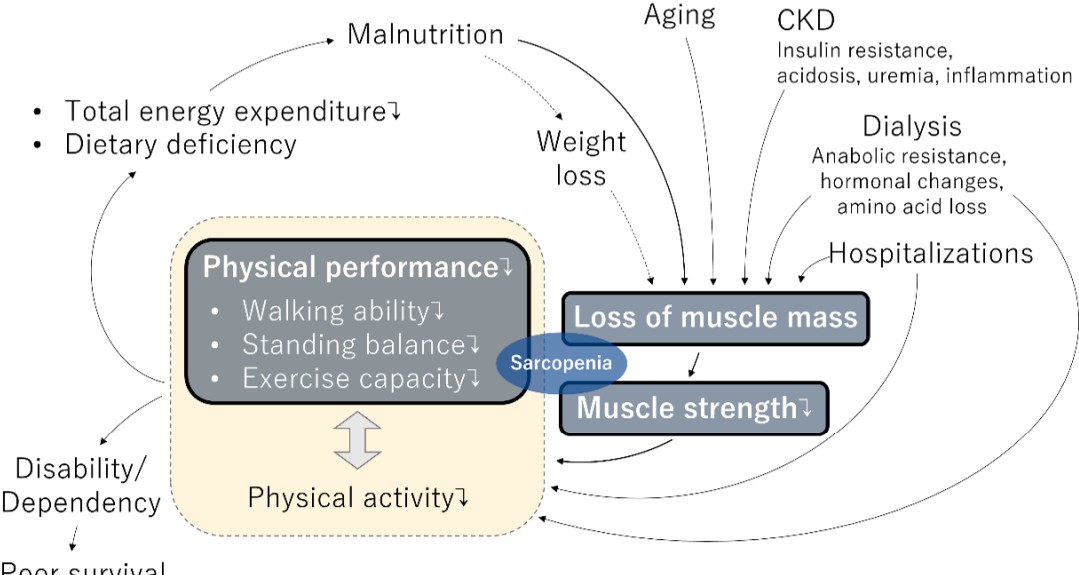

**Figure 1.** Cycle of frailty in patients with chronic kidney disease (CKD), adapted from Matsuzawa et al. [18].

### 3. Sarcopenia in the Cycle of Frailty

Sarcopenia has been described as a geriatric syndrome characterized by the loss of skeletal muscle mass and decreased muscle function and is at the center of the cycle of frailty. Patients with end-stage renal disease are at an increased risk of developing sarcopenia due to various factors [19–21,26]. Sarcopenia is diagnosed when patients have low muscle mass and low muscle strength or low physical performance according to the criteria of the Asian Working Group for Sarcopenia 2019 (AWGS2) [27]. The diagnostic criteria proposed by different working groups are summarized in Table 2 [27–30]. According to a recently performed meta-analysis, the prevalence of sarcopenia in patients on dialysis treatment was 28.5% [31], which is extremely high compared to rates ranging from 10.0 to 14.1% in the general population [32,33]. We examined the proportion of patients with sarcopenia according to the AWGS2 criteria among older Japanese patients on hemodialysis from three dialysis facilities and found that approximately 40% had been diagnosed with sarcopenia, and one in two aged 65 years and older had sarcopenia (Figure 2) [34]. Troublingly, sarcopenia ranges from asymptomatic to life-threatening, [35] and silent sarcopenia needs to be identified early via objective assessments. A diagnosis is essential for clinical management and therapeutic decision-making; however, it is not feasible in routine clinical practice for populations undergoing hemodialysis because it is time-consuming and resources are limited. Therefore, a convenient, objective, and rapid screening tool that can be used in clinical practice for hemodialysis patients whose physical conditions are dramatically altered, even for a short period, is needed. We recently conducted a study aimed to evaluate the ability of simplified tools to detect sarcopenia among patients on hemodialysis [34]. Our findings demonstrate that calf circumference and the creatinine-derived index could be considered as alternative means of discriminating sarcopenia in hemodialysis patients (Figure 3) [34]. The calf circumference was measured using a non-elastic tape at the point of the largest circumference. Both legs were measured, and the maximum value of both calves was used [34]. The creatinine-derived index, a disease-specific measure identifying low muscle mass and poor physical function [36,37], was calculated using the following formula:

**Table 2.** Diagnostic criteria of sarcopenia proposed by different working groups.

| Working Group | (A) Low Muscle Mass | (B) Low Muscle Strength | (C) Low Physical Performance | Diagnosis |
|---|---|---|---|---|
| IWGS (2011) [28] | ASM/height$^2$ (DXA): men $\leq$7.23 kg/m$^2$, women $\leq$5.67 kg/m$^2$ | - | Gait speed: <1.0 m/s | • Sarcopenia: (C) and (A) |
| EWGSOP2 (2019) [29] | ASM (BIA or DXA): men <20 kg, women <15 kg or ASM/height$^2$ (BIA or DXA): men <7.0 kg/m$^2$, women <6.0 kg/m$^2$ | Handgrip strength: men <27 kg, women <16 kg or five-time chair stand time: >15 s | Gait speed: $\leq$0.8 m/s or SPPB: $\leq$8 points or timed up and go test: $\geq$20 s or 400 m walk test: non-completion or $\geq$6 min for completion | • Sarcopenia probable: (B)<br>• Sarcopenia: (B) and (A)<br>• Sarcopenia severe: (B), (A) and (C) |
| AWGS (2020) [27] | ASM/height$^2$ (BIA): men <7.0 kg/m$^2$, women <5.7 kg/m$^2$ or ASM/height$^2$ (DXA): men <7.0 kg/m$^2$, women <5.4 kg/m$^2$ | Handgrip strength: men <28 kg, women <18 kg | Gait speed: <1.0 m/s or SPPB: $\leq$9 points or five-time chair stand time: $\geq$12 s | • Sarcopenia: (A) and (B) or (A) and (C)<br>• Sarcopenia severe: (A), (B) and (C) |
| ISPRM (2021) [30] | STAR (ultrasound): men <1.4, women <1.0 | Handgrip strength: men <32 kg, women <19 kg or five-time chair stand time: $\geq$12 s | Rise from a chair: inability or gait speed: $\leq$0.8 m/s | • Sarcopenia: (B) and (A)<br>• Sarcopenia severe: (B), (A) and (C) |

IWGS: the International Working Group on Sarcopenia; EWGSOP: the European Working Group on Sarcopenia in Older People; AWGS: the Asian Working Group for Sarcopenia; ISPRM: the International Society of Physical and Rehabilitation Medicine; ASM: appendicular skeletal muscle mass; DXA: dual-energy X-ray absorptiometry; BIA: bioimpedance; STAR: sonographic anterior thigh ratio; SPPB:, short physical performance battery.

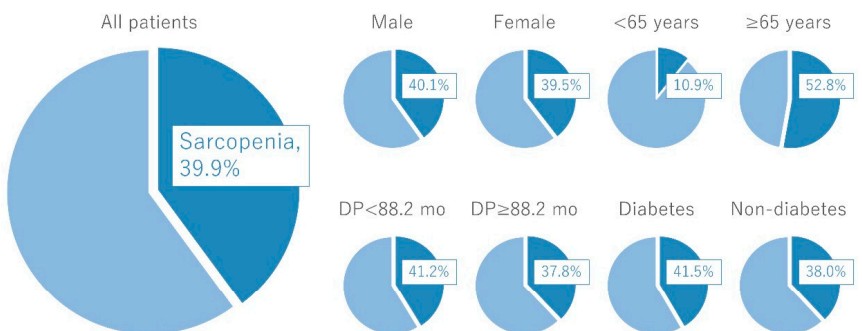

**Figure 2.** Prevalence rate of sarcopenia in patients undergoing hemodialysis, created based on a reference from Kakita et al. [34]. DP: dialysis period.

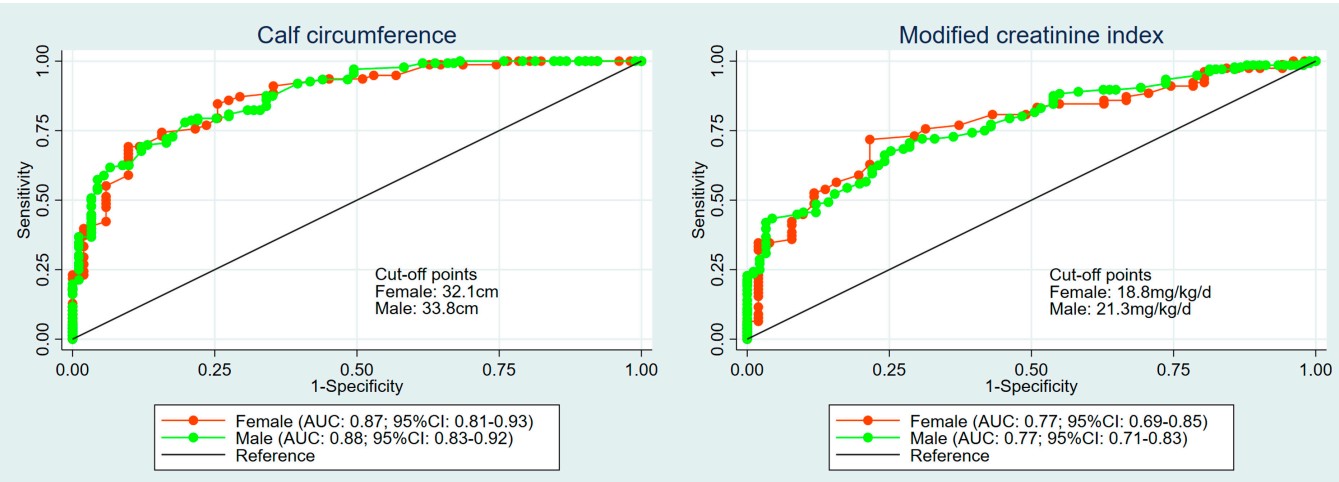

**Figure 3.** ROC curves of the simplified discriminant parameters against sarcopenia, created based on a reference from Kakita et al. [34].

Modified creatinine index (mg/kg/d) = 16.21 + 1.12 × (1 for men; 0 for women) − 0.06 × age (year) − 0.08 × single-pool Kt/V urea + 0.009 × serum creatinine level before dialysis (mmol/L) [36].

These are objective indicators that are easy to apply in clinical practice. Adding these assessments into routine clinical care will be valuable, not only for diagnosing sarcopenia, but also for improving prognostic stratification in patients on hemodialysis.

## 4. Management of Physical Frailty

In 2016, the European Renal Best Practice guideline development group underscored the importance of managing older patients with chronic kidney disease, especially with a routine assessment of physical function and activity [5]. An early identification of poor physical function and activity is essential for establishing a comprehensive management plan for patients on hemodialysis. We recommended a clinical physical frailty management algorithm for patients who require hemodialysis (Figure 4) [38], a modified version of Roshanravan's algorithm [39] tailored to Japanese populations. It consists of understanding a patient's physical function and activity level and exercise interventions. For low-functioning or sedentary patients who had been screened by routine evaluation, we encouraged participation in a supervised or home-based exercise program. We had previously evaluated the effects of a management program for physical frailty that consisted of the routine assessment of physical function and activity with feedback on the results of all-cause mortality and cardiovascular events for hemodialysis patients [40]. As a result, a lower proportion of program attendance was strongly associated with increased risks of mortality and cardiovascular events compared to those who attended the program more regularly. These results emphasize the importance of managing physical function and activity as part of routine clinical care.

Poor physical functioning, especially leg muscle strength, is common among hemodialysis patients and is strongly associated with decreased walking ability and lower basic and instrumental daily activity [41]. We previously evaluated thee lower extremity muscle strength using a handheld dynamometer in 190 clinically stable hemodialysis patients who did not require walking assistance. Approximately half of them had muscle strength below the cutoff value [42], which is used to determine whether patients can walk independently. This finding indicated that decreased muscle strength was already present. Given that lower extremity muscle strength correlated strongly and positively with gait speed and standing balance function [42], maintaining leg muscle strength seems to be a critical factor for preventing falls, fall-related fractures, and bedridden status. Fortunately, hemodialysis patients can improve poor muscle strength by resistance training. Low-intensity training

with ankle weights during dialysis sessions was shown to improve leg muscle strength and functional status in older patients [12].

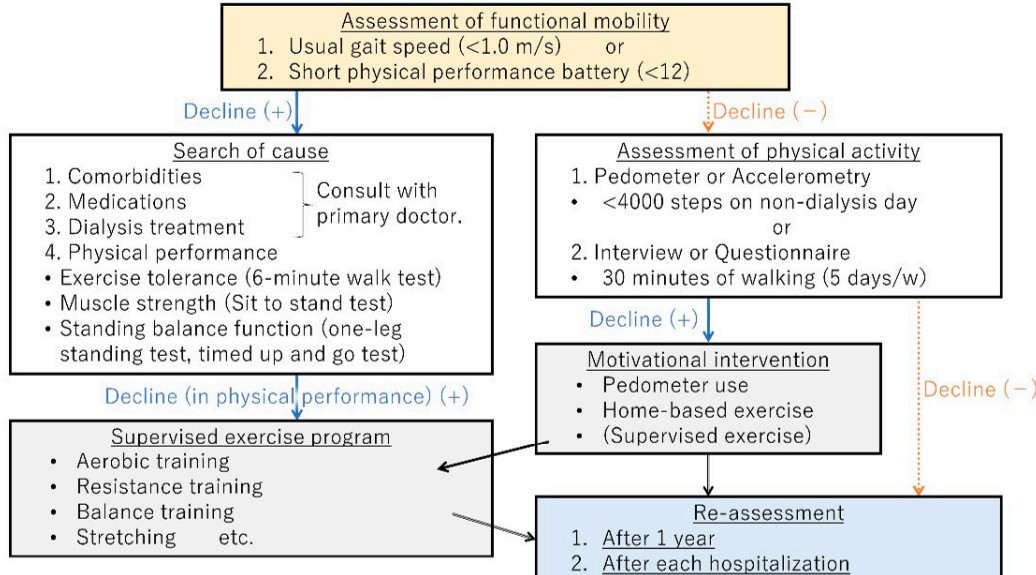

**Figure 4.** Clinical algorithm for the management of physical frailty in patients on hemodialysis, adapted from Matsuzawa et al. [18].

A systematic review and meta-analysis were previously conducted to evaluate the physical activity levels in patients with chronic kidney disease at different stages. The daily step counts in pre-dialysis patients, patients on peritoneal dialysis, patients on hemodialysis, and kidney transplant recipients were 5638 steps/day, 4264 steps/day, 4112 steps/day, and 8690 steps/day, respectively [43]. Patients on hemodialysis remained substantially less active as with patients on peritoneal dialysis and decreased physical activity assessed by questionnaire [44–48] or accelerometer-based methods [49,50] was strongly associated with higher mortality among patients on maintenance hemodialysis therapy. Goal setting is well-known to be a key motivational factor for increasing physical activity and is essential for successful intervention. We proposed walking 4000 steps per non-dialysis day as an initial minimum requirement for patients who require no assistance in walking [50]. This is a realistic goal for older adults that is consistent with the recommendations of the American College of Sports Medicine [51]. In addition, a decline in physical activity of >30% over the previous year was observed in almost one-quarter of patients undergoing hemodialysis and was associated with an elevated mortality risk independent of patient characteristics and baseline activity level [52]. On the other hand, we recently revealed that a lower physical activity level on "dialysis days" was also associated with higher risks of cardiovascular events and all-cause mortality independent to that on non-dialysis days [53]. Physical activity on dialysis days was restricted due to large fluctuations in vital signs during treatment or symptoms such as fatigue [54], so it is necessary to investigate the cause of decreased physical activity on dialysis days and to consider whether intradialytic exercise could be safely performed to cover the shortfall.

## 5. Frailty and Renal Transplantation

A previous prospective cohort study revealed that kidney transplant candidates with frailty were 38% less likely to be listed for kidney transplantation, had a 1.7-fold higher risk of waitlist mortality, and were 32% less likely to undergo kidney transplantation compared with non-frail individuals, even after adjusting for baseline characteristics including age, body mass index, and cause of end-stage renal disease [23]. Although the number of preemptive kidney transplants (PEKT) in Japan has increased recently, 60% of living kidney transplant recipients and 85% of cadaveric kidney transplant recipients still experience

dialysis therapy before transplantation and the average durations of dialysis were 2.6 years and 13.9 years, respectively. Therefore, for successful renal transplantation in older people, routine management of physical frailty in patients undergoing dialysis is essential. Additionally, exercise interventions against physical frailty need to be considered after kidney transplantation because that alone does not improve poor walking ability, low muscle function, or muscle loss.

## 6. Exercise Intervention after Kidney Transplantation

A recent meta-analysis showed that exercise therapy for patients with kidney transplant improved physical performance and quality of life [55], but most of this evidence targeted chronic, stable patients after a certain period of time had passed since transplantation. There have been few studies that have investigated exercise therapy in the early post-kidney transplantation period.

We started an early-phase exercise program for renal transplant patients, and after discharge from hospital, we reported its effect on physical performance and activity, quality of life, and kidney function [56]. Our program consisted of supervised aerobic training by physical therapists and physical activity in the ward and at home (Figure 5). Patients started the program on day 6 after transplantation. Supervised aerobic training was conducted during hospitalization and was performed for 3–4 weeks until discharge. Participants attended one or two sessions of supervised structured aerobic training per day at a rehabilitation center in the hospital 5 days/week. Aerobic training consisted of a 35–45 min/session on a treadmill walking or cycle ergometer exercise including a warmup and cool-down [56]. Exercise intensity was prescribed for patients at a 13–15 rating of perceived exertion on the Borg scale. Exercise therapy was stopped or the intensity was changed according to the criteria in Table 3. Physical activity instruction was provided for up to two months after kidney transplantation. On day 6 after transplantation, participants were instructed to wear a pedometer while performing a walking exercise. The following progressive target values were a guide: 3000 steps/day in the first week, 5000 in the second, and 5000 plus stair climbing movement in the third. The number of steps was confirmed by a physiotherapist during supervised aerobic training, and the target step count was corrected as necessary. As a result, exercise capacity and lower extremity muscle strength at 2 months after kidney transplantation were significantly higher in the exercise group than in the control. Regarding kidney function, in both the control and exercise groups, all patients succeeded in withdrawing from, or avoiding dialysis therapy, and there was no significant difference in the recovery curve of kidney function.

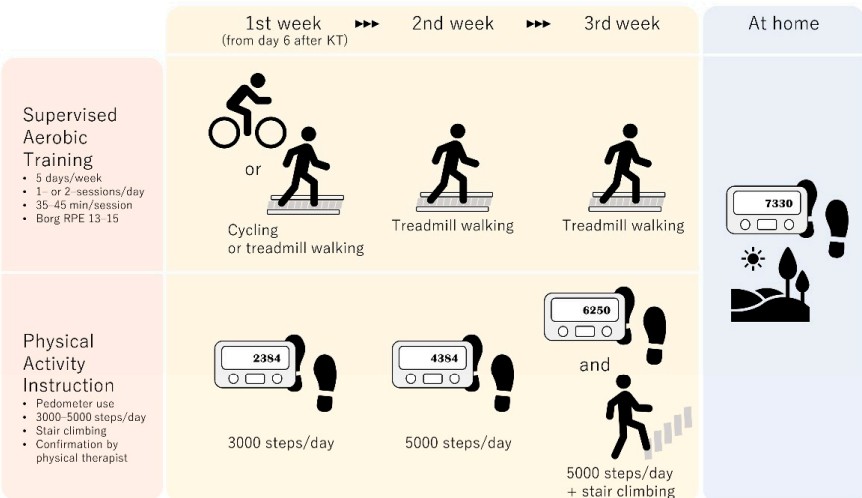

**Figure 5.** Protocol exercise intervention for patients after kidney transplantation, created based on a reference from Yamamoto et al. [56]. KT: kidney transplantation; RPE: rating of perceived exertion.

**Table 3.** Key points to note for safe implementation of exercise interventions among patients after kidney transplantation, created based on a reference from Yamamoto S. et al. [56].

| |
|---|
| **In the following cases, exercise therapy should be stopped or exercise intensity should be changed:** |
| (1)  Temperature higher than 38 °C. |
| (2)  Surgical wound pain higher than 7/10 on the visual analog scale. |
| (3)  Increase in the serum creatinine value for more than two days. |
| (4)  Increase in the serum creatinine value by 30% or more compared with the previous day. |
| (5)  Severe anemia (serum hemoglobin <7.0 g/dL). |
| (6)  Abstinence from eating. |
| (7)  Patients judged unsuitable for exercise therapy by the responsible nurses or the primary doctor for reasons other than the ones above. |

## 7. Summary

The mortality rates among patients on hemodialysis remain high, and one of the potential contributors might be the high burden of functional dependence, caused by impaired mobility, poor physical functioning, and muscle weakness—-the main components of physical frailty. As recent evidence has revealed, physical frailty can constitute a limiting factor for successful transplantation in patients with end-stage renal disease with or without hemodialysis. In addition, frailty among patients after transplantation has been identified as a risk factor for poor health outcomes. Therefore, initiating an exercise program to reduce or prevent physical frailty during the post-transplantation phase could be necessary. Although the concept of renal rehabilitation has become widely known among kidney health providers in recent years, kidney health professionals should make serious efforts to manage sarcopenia and physical frailty based on the evaluations of physical function and activity as well as exercise therapy.

**Author Contributions:** Conceptualization, R.M.; Formal analysis, R.M. and D.K.; Writing—original draft preparation, R.M.; Writing—review and editing, D.K.; Visualization, D.K. All authors have read and agreed to the published version of the manuscript.

**Funding:** This work was supported by research funding from Hyogo Medical University, Japan (R.M.) and the JSPS KAKENHI: 20K19332, Japan (R.M.).

**Institutional Review Board Statement:** No ethical approval was required because this study did not include confidential personal data and did not involve patient intervention.

**Informed Consent Statement:** Not applicable.

**Data Availability Statement:** Not applicable.

**Conflicts of Interest:** The authors declare no conflict of interest.

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
