# Peer review of "Renal Rehabilitation—Its Theory and Clinical Application to Patients Undergoing Daily Dialysis Therapy"

_kidneydial, doi:10.3390/kidneydial2040051_

Round 1

Reviewer 1 Report

Thank you for the opportunity to review this interesting manuscript. The paper is well-organized and well-written and the authors are to be congratulated for their efforts to review this inadequately studied area of renal medicine. 

I would suggest a few changes to improve readability. In the introduction, the authors describe differences in aging patterns in adults receiving hemodialysis in Japan compared with Europe and the US. They note that the population in Japan has aged more rapidly than in other countries and cite specific data for the Japanese population. It would be helpful to provide data for the other populations and to note that patients in Japan were younger than in other countries prior to 1990.

In section 2, "Functional Status and Physical Frailty", the authors refer to sarcopenia but only define it later in the paper. It would be better to include the definition with this earlier use of the term.

In section 4, "Management of Physical Frailty", the authors describe their treatment algorithm and note lower mortality and fewer cardiovascular events among those who attended the program regularly. It would be interesting to note if there were changes in the measurements of frailty described in the prior section 3, "Sarcopenia in the cycle of frailty".

Author Response

Response to the comments from the reviewer

No. kidneydial-1934476

Thank you for your thoughtful comments on our manuscript entitled, “Renal Rehabilitation—Its Theory and Clinical Application to Patients Undergoing Daily Dialysis Therapy”. We have tried to make full responses to all of the reviewers’ comments. The revised manuscript is substantially improved. I hope our revised manuscript meets the criteria for the publication in your most esteemed journal, Kidney and Dialysis. The modified parts of the manuscript are in red with underline.  Please confirm the attached file for details.

Reviewer 2 Report

 I recommend to add a description of patients with peritoneal dialysis (PD) if possible. Due to the characteristics of PD, it seems that the activity of PD patients is polarized between those who are likely to play a active social role and those who are home care and with low activity. Previous reports by other researchers are acceptable in this manuscript.

Line19 It is difficult to understand the intention of this sentence. The awareness of medical staff in renal disease treatment is progressing, but it can be said that the medical system is not well prepared. It seems that there is a shortage of specialists who can correctly evaluate the functions of patients, let alone kidney disease specialists.

It was thought that this would lead to the explanation of scoring in the text.

Line 75 Early stage of kidney disease were not excluded in this population in Ref.17. The description in the text is misleading.

Line 192

The relationship between the phenomenon of reduced chances of transplantation in Ref.23 and frailty may be included the causal and outcome relationship.  I have concern this description induce misunderstood whether diabetes and BMI are included in the multivariate in the study. I recommend to describe more detail about frailty risk in non-elderly ESKD to help our understanding.

Author Response

Response to the comments from the reviewer

No. kidneydial-1934476

Thank you for your thoughtful comments on our manuscript entitled, “Renal Rehabilitation—Its Theory and Clinical Application to Patients Undergoing Daily Dialysis Therapy”. We have tried to make full responses to all of the reviewers’ comments. The revised manuscript is substantially improved. I hope our revised manuscript meets the criteria for the publication in your most esteemed journal, Kidney and Dialysis. The modified parts of the manuscript are in red with underline.  Please confirm an attached file for details.

Reviewer 3 Report

The authors reviewed the theory and clinical application of renal rehabilitation for patients undergoing daily hemodialysis and renal transplantation. I would like to request some modifications as follows.

Major points

#1. The authors described that “most patients suffering from end-stage renal disease still undergo long-term dialysis therapy before transplantation.” It is true that period of dialysis therapy in cadaveric transplantation was long; however, 75% of living kidney transplant recipients who were majority compared with cadaveric kidney transplant recipients in Japan received dialysis therapy less than only 3 years, and about 50% of living kidney transplant recipients only received dialysis therapy less than only 1 year. The authors should revise the sentences according to the facts in this paragraph.

#2. A recent meta-analysis showed that exercise therapy for patients with kidney transplant improved physical performance and quality of life. The authors showed that an early-phase exercise program for renal transplant patients improved physical performance and activity early after surgery. On the other hand, this study could not demonstrate significant difference in the recovery of transplanted kidney function. Moreover, this article did not evaluate whether the clinical effect of early-phase exercise program would be maintained long after kidney transplantation, so that this study in not sufficient to describe that “initiating an exercise program to reduce or prevent physical frailty in the early post-transplantation phase is necessary” in summary.  

Author Response

(The authors gave the same response as above.)

Round 2

Reviewer 2 Report

The manuscript has been revised well.

Reviewer 3 Report

I am satisfied with revised manuscript in response to my review comments .
Thanks.